# Treatment of Brain Metastases of Non-Small Cell Lung Carcinoma

**DOI:** 10.3390/ijms22020593

**Published:** 2021-01-08

**Authors:** Agnieszka Rybarczyk-Kasiuchnicz, Rodryg Ramlau, Katarzyna Stencel

**Affiliations:** Department of Chemotherapy, Poznan University of Medical Sciences, Clinical Hospital of Lord Transfiguration, 61-848 Poznan, Poland; rramlau@gmail.com (R.R.); k.stencel@post.pl (K.S.)

**Keywords:** brain metastases, treatment, non-small cell lung carcinoma, EGFR, ALK, immunotherapy

## Abstract

Lung cancer is one of the most common malignant neoplasms. As a result of the disease’s progression, patients may develop metastases to the central nervous system. The prognosis in this location is unfavorable; untreated metastatic lesions may lead to death within one to two months. Existing therapies—neurosurgery and radiation therapy—do not improve the prognosis for every patient. The discovery of Epidermal Growth Factor Receptor (EGFR)—activating mutations and Anaplastic Lymphoma Kinase (ALK) rearrangements in patients with non-small cell lung adenocarcinoma has allowed for the introduction of small-molecule tyrosine kinase inhibitors to the treatment of advanced-stage patients. The Epidermal Growth Factor Receptor (EGFR) is a transmembrane protein with tyrosine kinase-dependent activity. EGFR is present in membranes of all epithelial cells. In physiological conditions, it plays an important role in the process of cell growth and proliferation. Binding the ligand to the EGFR causes its dimerization and the activation of the intracellular signaling cascade. Signal transduction involves the activation of MAPK, AKT, and JNK, resulting in DNA synthesis and cell proliferation. In cancer cells, binding the ligand to the EGFR also leads to its dimerization and transduction of the signal to the cell interior. It has been demonstrated that activating mutations in the gene for EGFR-exon19 (deletion), L858R point mutation in exon 21, and mutation in exon 20 results in cancer cell proliferation. Continuous stimulation of the receptor inhibits apoptosis, stimulates invasion, intensifies angiogenesis, and facilitates the formation of distant metastases. As a consequence, the cancer progresses. These activating gene mutations for the EGFR are present in 10–20% of lung adenocarcinomas. Approximately 3–7% of patients with lung adenocarcinoma have the echinoderm microtubule-associated protein-like 4 (EML4)/ALK fusion gene. The fusion of the two genes EML4 and ALK results in a fusion gene that activates the intracellular signaling pathway, stimulates the proliferation of tumor cells, and inhibits apoptosis. A new group of drugs—small-molecule tyrosine kinase inhibitors—has been developed; the first generation includes gefitinib and erlotinib and the ALK inhibitor crizotinib. These drugs reversibly block the EGFR by stopping the signal transmission to the cell. The second-generation tyrosine kinase inhibitor (TKI) afatinib or ALK inhibitor alectinib block the receptor irreversibly. Clinical trials with TKI in patients with non-small cell lung adenocarcinoma with central nervous system (CNS) metastases have shown prolonged, progression-free survival, a high percentage of objective responses, and improved quality of life. Resistance to treatment with this group of drugs emerging during TKI therapy is the basis for the detection of resistance mutations. The T790M mutation, present in exon 20 of the EGFR gene, is detected in patients treated with first- and second-generation TKI and is overcome by Osimertinib, a third-generation TKI. The I117N resistance mutation in patients with the ALK mutation treated with alectinib is overcome by ceritinib. In this way, sequential therapy ensures the continuity of treatment. In patients with CNS metastases, attempts are made to simultaneously administer radiation therapy and tyrosine kinase inhibitors. Patients with lung adenocarcinoma with CNS metastases, without activating EGFR mutation and without ALK rearrangement, benefit from immunotherapy. This therapeutic option blocks the PD-1 receptor on the surface of T or B lymphocytes or PD-L1 located on cancer cells with an applicable antibody. Based on clinical trials, pembrolizumab and all antibodies are included in the treatment of non-small cell lung carcinoma with CNS metastases.

## 1. Introduction

Lung cancer is one of the most common malignant neoplasms and the main cause of death from malignant neoplasms in Poland. Each year, more than 22,000 new lung cancer cases are recorded [1].

Metastases of lung cancer to the brain occur in 18–61% of patients [2,3,4]. Improving the effectiveness of oncological treatment leads to a higher survival rate but also increases the population of patients at risk of this complication [5,6]. Metastases to the central nervous system (CNS) are different from other metastases due to the occurrence of neurological disorders and often require discontinuing systemic treatment in order to carry out palliative care [7].

## 2. The Mechanism of Brain Metastasis Formation Is Similar to Other Organ Locations

As a result of mutations in cancer cells, the degree of invasiveness increases [8,9] (Figure 1). The cells detach themselves from the primary tumor and penetrate the blood vessels, reaching other organs through the bloodstream. Being a very well-vascularized organ, the brain is often subjected to metastases [10]. Metastatic cells arrest at distinct sites and extravasate through vascular walls into the brain parenchyma. Cancer cells proliferate at the metastatic niche, form colonies in this parenchyma, and the subsequent proliferation of cells leads to clinically detectable metastatic lesions [8,9].

Untreated metastases to the CNS lead to a gradual deterioration of the patient’s performance and to death within one to two months as a result of increased intracranial pressure. Avoiding or delaying these complications requires expertise in the radical and adjunctive treatment of brain metastases [11].

## 3. Non-Small Cell Lung Carcinoma without Activating the Mutation of Epidermal Growth Factor Receptor (EGFR) or Anaplastic Lymphoma Kinase (ALK)

A treatment method is chosen for lung carcinoma patients with CNS metastases based on their prognosis. It is determined by Radiation Therapy Oncology Group recursive partitioning analysis—RPA [12] (Table 1).

The eligibility criteria are the patient’s age, general performance, and the presence of metastases outside the CNS. The first group (class I) includes patients in good general condition, with a KI (Karnofsky Index) of 70% or more, less than 65 years old, without extracerebral metastases, and with good primary tumor control. The third class are patients in poor general condition, with a KI below 70%. The second class includes the remaining patients. In class III, radiotherapy of metastases to the CNS is not recommended due to a very bad prognosis. It is optimal to implement the best adjunctive treatment. The average period of survival is about two months [12].

In the case of a single metastasis to the brain up to four metastatic lesions to the CNS, either surgical removal or stereotactic radiotherapy (SRS) is recommended by the RPA in class I or II patients [7,13].

There is no evidence that the addition of whole-brain radiation therapy (WBRT) to stereotactic radiotherapy or surgery affects the overall survival of patients [14].

Data from a prospective study by Japanese researchers (JLGK0901) indicates that stereotaxis may be relevant in patients with more than three CNS metastases [15]. The observational trial lasted three years and included 1194 patients [15] with 1–10 newly diagnosed CNS metastases. The largest tumor volume was <10 mL and <3 cm in the longest dimension. The total cumulative volume did not exceed 15 mL, and the Karnofsky performance status was 70% or higher. All patients qualified in this way received stereotactic CNS radiotherapy. The overall survival (OS) of patients after stereotaxis was 13.9 months in 455 patients with a single metastasis, 10.8 months in 531 patients with two–four metastatic lesions, and 10.8 months in 208 patients with 5–10 metastatic lesions. An equal overall survival (OS) in patients with two–four metastatic lesions and 5–10 lesions indicates that stereotactic radiosurgery (SRS) is an important alternative to whole-brain radiotherapy in selected patients in good general condition.

Stereotaxis with or without whole-brain radiotherapy was analyzed in a third phase trial in which patients with one–four brain metastases of lung carcinoma were randomized [16]. Three hundred and sixty-four patients meeting the criteria for inclusion in the trial were analyzed. Fifty-one percent of patients received only stereotactic radiotherapy, while 49% received SRS followed by whole-brain radiation therapy (WBRT). It was shown that the age of patients significantly affects their survival. Stereotactic radiotherapy as a stand-alone treatment improves survival in patients aged 50 years or younger, with no difference in the age group over 50 years. Patients with a single metastasis experienced significantly longer survival than patients with two–four metastases. In the assessment of cognitive disorders during treatment, patients under 50 years of age tolerated the therapy better in both arms of the trial. Patients with a single CNS metastasis, compared to patients with two–four lesions, had less severe cognitive impairment. Local disease control was better in the arm with SRS plus WBRT in both age groups.

In the third-phase QUARTZ trial [17], the role of whole-brain radiation therapy (WBRT) was assessed in patients with non-small cell lung carcinoma with inoperable CNS metastases. The patients were randomized into two groups. In one group, they received radiation therapy—WBRT 20 Gy in four fractions—and steroid therapy, and in the other group, the best adjunctive treatment without radiotherapy. The mean survival duration of patients in the radiotherapy arm was 49 days and 51 days in the optimal adjunctive treatment arm. In both groups, there were no differences in the quality of life and the use of steroids. The entire brain can be subjected to radiation therapy in a 20-Gy regimen in five fractions or 30 Gy in 10 fractions [18]. Alternative fractionation: 40 Gy in 20 fractions twice a day does not affect patients’ survival times. Attempts have been made to use chemotherapy as a radiosensitizer without improving patients’ survival times [19].

In patients with asymptomatic CNS metastases who have not yet received systemic treatment, the therapy sequence should be considered. In a study published in 2014 [20], patients with non-small cell lung carcinoma with asymptomatic CNS metastases (one to four lesions) received either stereotactic radiosurgery (SRS) followed by a two-drug cisplatin-based chemotherapy or chemotherapy alone. The average age of the patients was 58 years, with a mean total survival time of 14.6 months in the arm with stereotactic radiotherapy and chemotherapy; in the arm with chemotherapy alone—15.3 months. The average time to progression in the CNS was 9.4 months in the arm with SRS; in the arm with chemotherapy alone—6.6 months. The symptomatic progression of CNS lesions was more frequently observed in patients without stereotactic radiotherapy [19].

In the phase 3 trial [21], patients with CNS metastases of non-small cell lung carcinoma (NSCLC) received chemotherapy—cisplatin with vinorelbine (days 1, 8, 15, and 22)—courses every 28 days, with a maximum of six courses. Whole-brain radiation therapy (WBRT)—30 Gy in 10 fractions—took place early in some patients—on days 1–12 of the first chemotherapy course and, in the second arm, after two chemotherapy courses (56 days). The objective response rate was 20% in the early radiotherapy arm and 21% in the delayed radiotherapy arm. The average survival duration in patients with delayed radiotherapy was 24 weeks and, in patients with early radiotherapy, 21 weeks. The results indicate that, during chemotherapy treatment, the implementation time of CNS palliative radiotherapy in patients with asymptomatic NSCLC brain metastases does not affect patients’ survival duration [21].

In patients with symptomatic metastases of lung cancer to the CNS, the recommended dose of corticosteroids used long term in the prevention of cerebral edema is 4 mg of dexamethasone daily. Increasing the dose of the steroid to 16 mg daily does not improve the disease control but generates treatment toxicity [22].

## 4. Non-Small Cell Lung Carcinoma with Present Epidermal Growth Factor Receptor (EGFR) Mutation and ALK Rearrangement

The discovery of the EGFR mutation [23,24] and ALK rearrangement [24] and then the introduction of first-generation tyrosine kinase inhibitors (gefitinib—Figure 2A and erlotinib—Figure 2B) to the treatment of non-small cell lung carcinoma allowed, compared to platinum-based dual-drug chemotherapy, for longer progression-free survival (PFS), higher objective response rates (ORR), and better disease control rates (DCR) in comparison with two-drug platinum-derivative-based chemotherapy [23,24,25].

EGFR is one of the four members [25] of the HER family receptors, which comprise [26] EGFR/HER1/erbB1, HER2/erbB2, HER3/erbB3, and HER4/erbB4 [27]. EGFR signaling [25] is triggered by the binding of growth factors, such as Epidermal Growth Factor (EGF) [25,28], resulting in the dimerization of EGFR molecules [27]. Autophosphorylation and transphosphorylation of the receptors through their tyrosine kinase domains [25] leads to the recruitment of downstream effectors and the activation of proliferative and cell survival signals [25]. In recent years, intensive research has been dedicated to the Epidermal Growth Factor Receptor (EGFR) [27] due to its significant role in the pathogenesis [27,29] of malignant tumors. In many types of cancers, intracellular pathways modulated by EGFR have been identified [25,28] as crucial factors influencing tumor survival and development [30]. On the other hand, EGFR has also been shown to be a promising molecular target [25,26,27] for potential therapeutic agents. Attempts to modify the signal transduction exerted by EGF have been made either by blocking [25] the activity of certain elements of the EGFR pathway or by direct inhibition of the EGF receptor itself [27]. Gefitinib and erlotinib target the ATP cleft [31] within the tyrosine kinase Epidermal Growth Factor Receptor (EGFR). Specific activating mutations within the tyrosine kinase domain of the EGFR molecularly correlate to the responses [23,24,25] to gefitinib or erlotinib (Figure 3A,B).

In Figure 3A, the inhibitor (dark blue), representing gefitinib, occupies the ATP cleft. The locations of the two missense mutations are shown within the activating loop of the tyrosine kinase (light blue); the three in-frame deletions are all present within another loop (shown in red), which flanks the ATP cleft. Figure 3B shows a close-up view of the EGFR tyrosine kinase domain, with the critical amino acids implicated in binding the inhibitor. Specifically, 4-anilinoquinazoline compounds such as gefitinib inhibit catalysis by occupying the ATP-binding site, where they form hydrogen bonds with methionine 769 (M769) and cysteine 751 (C751) residues, whereas their anilino ring is close to the methionine 742 (M742), lysine 721 (K721), and leucine 764 (L764) residues (all shown in green). In-frame deletions within the loop that is targeted by mutations (shown in red) are predicted to alter the positions of these amino acids relative to that of the inhibitor. Mutated residues (red) are shown within the activation loop of the tyrosine kinase (light blue).

Gain-of-function mutations [32] in the tyrosine kinase domain of the EGFR gene markedly increase the sensitivity to EGFR tyrosine kinase inhibitors (TKIs) [33]. It has been shown that 10–30% of all lung adenocarcinomas [34,35] contain an EGFR-activating mutation. EGFR mutations occur mostly in adenocarcinoma, younger women and girls [26], and never-smokers [23,24,25]. The increased prevalence [32] of EGFR mutations in the metastatic disease (early stage—14, 2% and metastatic—30, 3%) in the dataset may partially reflect referral bias [26] (Figure 4).

The most common oncogenic mutations are deletion in exon 19 (45–50% of all somatic EGFR mutations) and a point mutation (L858R) in exon 21 (35–45% of mutations) [25,36,37]. Ex20Ins mutations are the third-most common EGFR-activating mutations in NSCLC [38], which collectively account for approximately 4% to 10% of all EGFR mutations [35]. These mutations are predictive of the clinical activity of the EGFR TKIs [39], which yield a superior RR (response rate) [39,40] and PFS [40,41], as well as a better QoL (quality of life) [39,40,41] scores when compared with combination chemotherapy in the first-line setting [23]. The discovery of EGFR mutations and ALK rearrangements also contributed to the development of a new scale [37] of prognostic factors in patients with brain metastases of non-small cell lung carcinoma, taking into account the presence of EGFR mutations or ALK rearrangements. The Lung Cancer Molecular Markers Graded Prognostic Assessment (Lung-molGPA) index facilitates making clinical decisions in this group of patients. In addition to the previous parameters [12], such as the patient’s age, general performance, presence or absence of cancer outside the CNS, the number of brain metastases (one–four or >four), it also takes into account the gene status of the EGFR and ALK mutations. The higher the number of points obtained on this scale, the better the prognosis and longer survival of patients [37] (Table 2).

First-generation tyrosine kinase inhibitors block [38,39] the EGFR receptor in a reversible manner. A better control of neoplastic disease during treatment with gefitinib or erlotinib [40,41], and the longer lives of patients, drew attention to the problem of metastatic lesions in the CNS. Lung cancer patients treated with first-generation TKI achieved a mean survival time of 33.1 months. After the diagnosis of disease progression in the CNS or in the meninges, the average survival time was 5.5 and 5.1 months. The incomplete penetration of drugs into the CNS through the blood–brain barrier causes a worse response to the first-generation TKI treatment in the brain and meninges [42]. Despite the low molecular weights of gefitinib and erlotinib, their penetration rates into the cerebrospinal fluid (1.13% and 2.77%, respectively) and the CNS concentration rates are low (3.7 ng/mL and 28.7 ng/mL, respectively) [43] (Table 3).

Attempts have been made to increase the doses of gefitinib or erlotinib [44] or to introduce the pulsatile administration of drugs in patients with metastatic lesions in the CNS. The achieved therapeutic effects were still unsatisfactory due to the fact that higher doses of the first-generation TKI [45] increased the drug concentration index in the CNS, but the obtained effect was short-lived. A prolonged administration of high doses of erlotinib or gefitinib causes unacceptable toxicity and is not used [44,45,46].

Afatinib (Figure 5) is a second-generation tyrosine kinase inhibitor.

Acquired resistance occurs [47] in patients who initially benefit from EGFR-targeted therapies (first-generation tyrosine kinase inhibitors) [25,26].

A clinical definition of acquired resistance to EGFR TKIs: acquired resistance in systemic progression (by Response Evaluation Criteria in Solid Tumors (RECIST) or World Health Organization (WHO) criteria) after a complete or partial response or >six months of stable disease after treatment with targeted therapy [48]. It irreversibly binds to the EGFR receptor and also has a higher affinity for the receptor compared to first-generation drugs. The studies LUX-Lung 3 [49] (cisplatin with pemetrexed) and LUX-Lung 6 [50] (cisplatin with gemcitabine) demonstrated the superiority of TKI over platinum-based two-drug chemotherapy with new-generation drugs. In the presented studies, patients receiving TKI compared to chemotherapy benefited from longer progression-free survival (PFS). They showed higher objective response rates (ORR) and a better disease control rate (DCR). The CNS penetration rate for afatinib is below 1%, and the CNS concentration is 0.46 ng/mL [51]. The LUX-Lung 3 and LUX-Lung 6 [49,50] studies were analyzed, taking into account asymptomatic brain metastases. The progression-free time in the LUX-Lung 3 trial [52] in patients with CNS metastases was 11.1 months in the afatinib arm and 5.4 months in the chemotherapy arm. In the LUX-Lung 6 trial, patients with CNS metastases treated with afatinib [53,54,55] achieved a progression-free time of 8.2 months, and in the chemotherapy arm, PFS was 4.7 months. Progression-free time in the afatinib arm compared to chemotherapy was equal in patients without brain metastases and in patients with CNS metastases [50,56,57]. The LUX-Lung 7 trial compared gefitinib with afatinib and included patients with central nervous system metastases. The mean follow-up was 27.3 months; progression-free survival for the afatinib arm was 11 months and 10.9 months for the gefitinib arm. The time to treatment failure for afatinib was 13.7 months and, for gefitinib, 11.5 months. Afatinib and gefitinib in the LUX-Lung 7 trial—no difference in the overall survival (OS) [58]. Brueckl et al. (ESMO 2018 Congress, abstract 1449P) [59] presented an analysis of GIDEON, a prospective noninterventional study that was conducted in Germany to investigate the activity and tolerability of first-line afatinib in routine clinical care. Among 151 treated patients, the majority (72.8%) started treatment at an afatinib dose of ≥40 mg; 61.8% of them had dose reductions. In the group of patients starting at <40 mg, 46.2% had dose reductions, while dose increases were performed in 33.3%. The safety profile of afatinib was consistent with the known safety profile identified by the clinical trials. In spite of relatively high proportions of patients with brain metastases (approximately 30%) and uncommon *EGFR* mutations (approximately 13%), the results corroborated the clinical data for afatinib in the routine setting. The median PFS was 12.9 months, with a 12-month PFS rate of 54.6%. Seventy-three percent of patients responded, and 90% obtained disease control. Both the ORRs and disease control rates (DCR) were independent of the type of *EGFR* mutation, the presence of baseline brain metastases, and the starting dose (Figure 6).

Osimertinib (Figure 7) is a third-generation tyrosine kinase inhibitor. In the AURA 3 clinical trial [60], it was compared to pemetrexed and cisplatin or carboplatin-based two-drug chemotherapy [61,62].

It was the second-line treatment for all patients, with the first- and second-generation EGFR TKI used in the first-line treatment. After the disease progressed, the T790M mutation determining the resistance [63] to drugs from the first- and second-generation TKI groups was determined, and patients were randomized to the Osimertinib arm or to the chemotherapy arm [64,65]. The trial also included patients with metastases to the central nervous system, without symptoms resulting from focal lesions in the CNS, who did not require treatment with steroids for at least four weeks before the start of the trial. The median treatment duration was 10.1 months for patients treated with Osimertinib (Osimertinib, *n* = 279) and 4.4 months for patients treated with chemotherapy (*n* = 140). The objective response rate (ORR) was 71% for the Osimertinib treatment and 31% for chemotherapy-treated patients [63,64]. A subgroup analysis was performed; patients with measurable CNS lesions (one or more brain lesions) were included in the first group and patients with one or more lesions measurable and nonmeasurable in the CNS in the second group. In the first group of patients [64,65], the ORR was 70% in the Osimertinib arm and 31% in the chemotherapy arm [66]. In the second group of patients [64,65,67], the ORR was 40% in the Osimertinib arm and 17% in the chemotherapy arm. In both groups of patients, the mean response time in the CNS was 8.9 months in the treatment with Osimertinib and 5.7 months in the treatment with chemotherapy [67,68]. The mean PFS in the group of patients with measurable changes in the treatment with Osimertinib was 11.7 months, while, in chemotherapy, it was 5.6 months [69].

The EGFR T790M mutation [70] is the most common mechanism of TKI first- and second-generation resistance (detected in 50–60% of patients) [25]. It is unlikely that any erlotinib combination [70,71,72] will overcome this specific drug resistance mechanism.

Osimertinib, a third-generation small molecule tyrosine kinase inhibitor, is recommended in patients with the T790M resistance mutation [66,67]. It is also effective in patients with metastases to the central nervous system and the meninges [67,73,74].

In the phase 3 FLAURA [71,72,73,74] clinical trial, patients receiving Osimertinib achieved a PFS of 18.9 months and 10.2 months in the control arm (gefitinib or erlotinib). Patients with CNS metastases also benefited from treatment with Osimertinib [74,75].

Osimertinib is a third-generation tyrosine kinase inhibitor and has demonstrated high tolerability [73,74,75]. Some patients showed resistance to this drug, and the major mutation site is C797S on the EGFR gene (discovery of genome sequencing) [75,76]. In the future, when EGFR TKI drug resistance occurs [75,76], genetic testing could be used to select the treatment method corresponding to the resistance mechanism [74,75,76].

Progress in the field of molecular biology in recent years has enabled the identification of potential oncogenic pathways [77,78]. In 2007, Soda and his colleagues found an echinoderm microtubule-associated protein-like 4 (EML4) ALK fusion gene from non-smell-cell lung cancers [77]. These ALK fusion proteins can induce the constitutive activation of the ALK tyrosine kinase [77,78]. The oligomerization of domains such as the coiled-coil [77] domain of the fusion partner gives stimulation [79] ALK downstream pathways as a result [79]. The P13K-AKT-Mtor, RAS-MAPK-ERK, or JAK-STAT pathways are constitutively activated [77,79].

ALK mutations are rare and can be found in approximately 3–7% of patients with the diagnosis of NSCLC [77,78,79]. ALK mutations are more common in young, nonsmoking men with adenocarcinoma [78,79].

Crizotinib is an ATP-competitive, orally bioavailable ALK inhibitor [80] and was first applied for the treatment of EML4 ALK-positive NSCLC [81]. Crizotinib (Figure 8) was introduced based on the phase 3 Profile 1014 study [81] as a standard of treatment in patients with ALK-positive lung cancer.

This first-generation tyrosine kinase inhibitor has a concentration rate in the cerebrospinal fluid of 0.616 ng/mL and a penetration rate to the cerebrospinal fluid of 0.26% [51,80]. In the Profile 1014 trial [81], crizotinib achieved significantly longer PFS compared to chemotherapy (nine months vs. four months), and after 12 and 24 weeks of treatment, higher intracerebral DCR of 85% and 65% was observed in the arm with crizotinib and 45% and 25% in the arm with chemotherapy. The intracerebral control of the disease was also better in patients with metastases to the CNS—23% compared to chemotherapy. However, the isolated progression of the disease in the CNS was more frequent during treatment with crizotinib; extracerebral progression was more frequent during the treatment with chemotherapy (pemetrexed cisplatin). Patients with untreated CNS metastases or progression [82,83] of the disease were not randomized for the trial, and 20% of patients participating in the trial had CNS radiotherapy.

Nearly one-third of patients treated with crizotinib had CNS metastases in the first year of therapy. In some of these patients, it was the only location of neoplastic disease progression [83,84].

Alectinib (Figure 9) is a second-generation tyrosine kinase inhibitor used in patients with ALK-positive lung tumors and is also effective in the central nervous system [85,86].

Alectinib shows a concentration level of 2.69 nM [87,88] in the cerebrospinal fluid, and the penetration rate into the cerebrospinal fluid is 86% [51,87,88].

The ALEX phase 3 [89] clinical trial included previously untreated patients with advanced ALK-positive lung cancer. The patients received crizotinib or alectinib [89,90]. The primary endpoint was progression-free survival. The mean follow-up was 17.6 months in the crizotinib arm and 18.6 months for alectinib. The progression-free time was significantly longer in patients treated with alectinib [89,90] and was 12 months, while, in the crizotinib arm, it was 8.5 months [89,90]. In this trial, alectinib demonstrated therapeutic efficacy in patients with metastases to the central nervous system. Twelve percent of patients treated with alectinib (18 patients) and 45% (68 patients) treated with crizotinib demonstrated changes in the CNS. The one-year (12 months) CNS cumulative events (progression) level was 9.4% vs. 41.4% when comparing alectinib with crizotinib. Good intracerebral disease control coexisted with PFS—a mean average of 25.7 months for alectinib [90,91] and 10.4 months for crizotinib. Treatment toxicity was also lower in patients receiving alectinib and had a favorable safety profile [90,91].

Another ALK inhibitor was found to overcome crizotinib resistance and to better control disease in the CNS [92]. Ceritinib (Figure 10) is a second-generation ALK TKI and has a cerebrospinal fluid penetration rate of 15% [51]. Ceritinib is effective in patients with the I117N resistance mutation [92,93,94].

The phase 3 clinical trial ASCEND-4 [92] compared ceritinib with chemotherapy as the first-line treatment in patients with advanced lung cancer and ALK rearrangement. In patients with metastases to the CNS, the mean PFS was 10.7 months in the ceritinib arm and 6.7 months in the chemotherapy arm [92]. The overall intracranial response rate in patients with measurable CNS changes at the baseline was 72.7% for ceritinib and 27.3% for chemotherapy [92,93,94].

In the phase 1 ASCEND-1 trial [95], ceritinib achieved a total intracerebral ORR of 36% in ALK TKI previously treated patients and 63% in ALK TKI untreated patients (patients had baseline CNS measurable changes). In the ASCEND-2 trial the intracerebral ORR was almost 40%, and the intracerebral DCR was 85% [95,96,97].

Brigatinib (Figure 11) is a second-generation ALK TKI. Brigatinib was shown to be active against the G1202R mutation [98]. The G1202R mutation is resistant to first- and second-generation ALK inhibitors (crizotinib, alectinib, and ceritinib) [98,99].

It was noted that the G1202 mutation was discovered in about 50% of relapse patients following the use of brigatinib [98,99]. Brigatinib, another second-generation ALK inhibitor, demonstrated substantial activity in patients with crizotinib refractory *ALK*-positive NSCLC; however, its activity in the alectinib refractory setting is unknown [98].

The phase 2 ALTA trial [100,101] evaluated the efficacy of brigatinib in patients with advanced ALK-positive non-small cell lung carcinoma previously treated with crizotinib [100,101]. Patients were randomized to two arms of the trial—in one arm, the dose was 90 mg, and, in the other arm, 180 mg for seven days, then 90 mg [101]. In patients with measurable changes in the CNS, the ORR at a higher dose of the drug was 67%, and, at a lower dose, 37%. The DCR exceeded 80% in both arms. In the case of nonmeasurable CNS metastases, the ORR and DCR were higher in patients receiving the higher dose of the drug (19% vs. 6% and 87% vs. 72%). Two-thirds of patients receiving the higher dose of the drug and having measurable lesions in the CNS had an intracerebral response lasting, on average, 16.6 months [100,101]. Brigatinib was compared with crizotinib in a phase 3 trial in patients with ALK-positive [101] lung cancer who had not been previously treated with TKI. Ninety patients had baseline CNS metastases, and 39 patients had measurable CNS lesions with a diameter >10 mm. The intracerebral response to treatment in patients with measurable lesions was 78% in the brigatinib arm and 29% in the crizotinib arm. In the brigatinib group, 9% of patients had disease progression in the CNS, and, in the crizotinib group, 19% of patients [99,100]. Twelve-month PFS in the group of patients with metastatic lesions in the CNS at baseline was higher in the brigatinib arm—67% than in the crizotinib arm—21% [101].

Lorlatinib (Figure 12) is a third-generation ALK inhibitor with a penetration rate to the CNS of 20–30% [51]. Lorlatinib is indicated for the treatment of patients with ALK-positive metastatic non-small cell lung cancer [102] whose disease progressed on crizotinib [103] and at least one other ALK inhibitor. Lorlatinib has been shown to be active against almost all of the previously identified ALK TKI resistance mutations, including G1202R [103,104]. It is supposed to overcome the resistance of cancer cells to early-generation drugs [104].

In a phase 1 trial [105], an intracerebral RR of 44% was achieved in the lorlatinib arm in patients with metastatic changes in the CNS for measurable and nonmeasurable lesions and 60% for measurable lesions. Approval was based on a phase 2 study [106,107] in which lorlatinib demonstrated a substantial overall and intracranial response [106,107].

## 5. Simultaneous CNS Radiotherapy and TKI Therapy

It was shown that lung cancer cells with the EGFR mutation are more radiosensitive [108] than those without. At the same time, lung cancer patients with EGFR mutations have a 50–70% risk of brain metastases [109]. Before the era of targeted lung cancer treatment, patients had either neurosurgical surgery, SRS, or whole-brain radiotherapy with the occurrence of metastases to the brain.

Two hundred and thirty patients with CNS metastases and EGFR mutations were identified and divided into two groups [110]. In one group, 116 patients received TKI (gefitinib, erlotinib, or icotinib), and, in the other group (51), TKI and simultaneous radiotherapy of the whole brain. An ORR of 52% was achieved in both groups; OS in the radiotherapy and TKI arm was 26.4 months and, for the treatment with only TKI, 21.6 months. Compared with TKIs alone, EGFR TKIs plus WBRT demonstrated intracranial progression-free survival (PFS) of 6.9 vs. 7.4 months (*p* = 0.232) and systemic PFS of 7.5 vs. 7.9 months (*p* = 0.546) [110].

In a meta-analysis of seven trials [109] involving 1086 patients with brain metastases, TKI therapy alone was compared with radiotherapy used before TKI therapy. It was shown that patients with non-small cell lung carcinoma and brain metastases who received radiotherapy prior to TKI therapy had longer intracerebral PFS and longer OS [109]. The analysis in the subgroups showed that the survival time of patients was longer in the group with one–three metastatic lesions [109], and shorter OS was obtained by patients with more metastatic lesions. The analysis confirmed that radiotherapy, by damaging the blood–brain barrier, increases the effectiveness of TKI therapy. Consequently, the combined therapy reduces relapse and improves the overall survival [108,109].

In patients with lung cancer and brain metastases, attempts were made to combine up-front CNS radiation and TKI therapy. Based [109] on the current available evidence, patients of non-small cell lung cancer with brain metastases and EGFR mutations have better OS and iPFS (intracerebral progression free survival) when they receive up-front radiotherapy and TKI than TKI alone [108,109,110].

The subgroup analysis [109] showed that never-smokers lived longer compared to tobacco smokers, and patients diagnosed with adenocarcinoma lived longer compared to other histopathological types. Patients with a better overall performance status (ECOG) lived longer than patients in worse general condition. In the group of patients with symptomatic brain metastases who received TKI and simultaneous whole-brain radiotherapy, significantly worse intracerebral PFS was observed compared to patients treated only with TKI [109,110,111].

Currently, it is not recommended to discontinue TKI therapy while radiating the whole brain. For stereotaxis (SRS), it is recommended to discontinue TKI three days before SRS and restart it three days after treatment; therefore, the interval is seven days [111].

## 6. Immunotherapy of Lung Cancer with Brain Metastases

Pembrolizumab is the drug of choice in the first-line treatment of patients with a PD-L1 expression in >50% of tumor cells in patients with non-small cell lung carcinoma without the EGFR or ALK mutation. In the registration trial of pembrolizumab—Keynote-024 [112], 9% of patients had CNS metastases, and, in Keynote-010 [113], 15% of patients had CNS metastases. Pembrolizumab is recommended for the first-line treatment of stage IV non-small cell lung carcinoma (including patients with stable metastatic lesions in the central nervous system) [114].

In the CheckMate 057 trial [115], nivolumab was administered to patients with non-squamous lung cancer as a second-line treatment. Patients achieved an OS of 12.2 months; in the arm with chemotherapy, the OS was 9.4 months. Patients with stable metastatic lesions in the central nervous system were randomized for the trial [115].

In the EAP (expanded access program) 1588 trial [116], nivolumab was administered to patients with the IIIB/IV tostages of non-squamous lung cancer after progression on prior systemic therapy. Four hundred and nine patients had CNS metastases. They were neurologically stable and could receive a steroid therapy of up to 10 mg of prednisone daily. In the group of patients with metastatic lesions in the brain, the mean follow-up time was 6.1 months (0.1–21.9); the DCR was 39%, and the mean OS was 8.6 months; the CNS disease stabilized in 96 patients, 64 patients achieved a partial response, and 4 complete CNS responses during the nivolumab treatment [116]. Currently, nivolumab is recommended for the second-line treatment of stage IV non-small cell lung carcinoma [114].

Lung cancer metastases to the central nervous system pose a serious problem in oncological treatment. These lesions not only cause progression of the neoplastic disease but also manifest focal symptoms from the CNS, affecting the general condition of patients and worsening contact with them. Neurosurgery, stereotactic radiotherapy, and chemotherapy help to improve the clinical conditions of patients. Introducing new molecules into clinical practice gives a chance not only to improve the general condition of patients but also to prolong their lives.

## Figures and Tables

**Figure 1 ijms-22-00593-f001:**
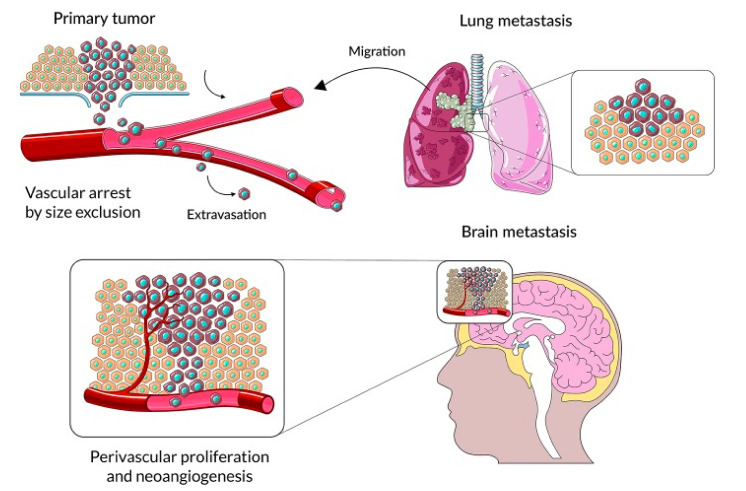
The main stages of cancer cell colonization of brain parenchyma (source: You H. et al., Front Immunol., 2019 [8]).

**Figure 2 ijms-22-00593-f002:**
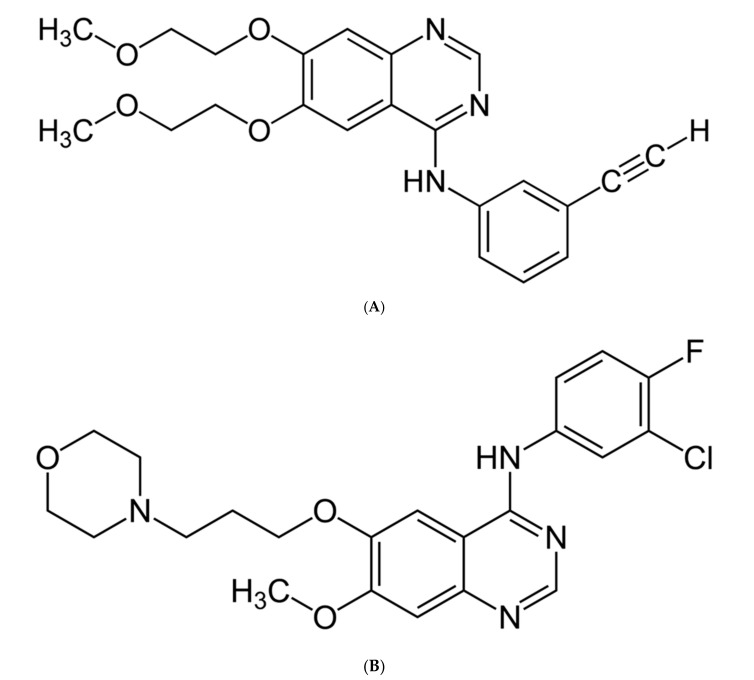
(**A**) Gefitinib—first-generation tyrosine kinase inhibitor; (**B**) Erlotinib—first-generation tyrosine kinase inhibitor.

**Figure 3 ijms-22-00593-f003:**
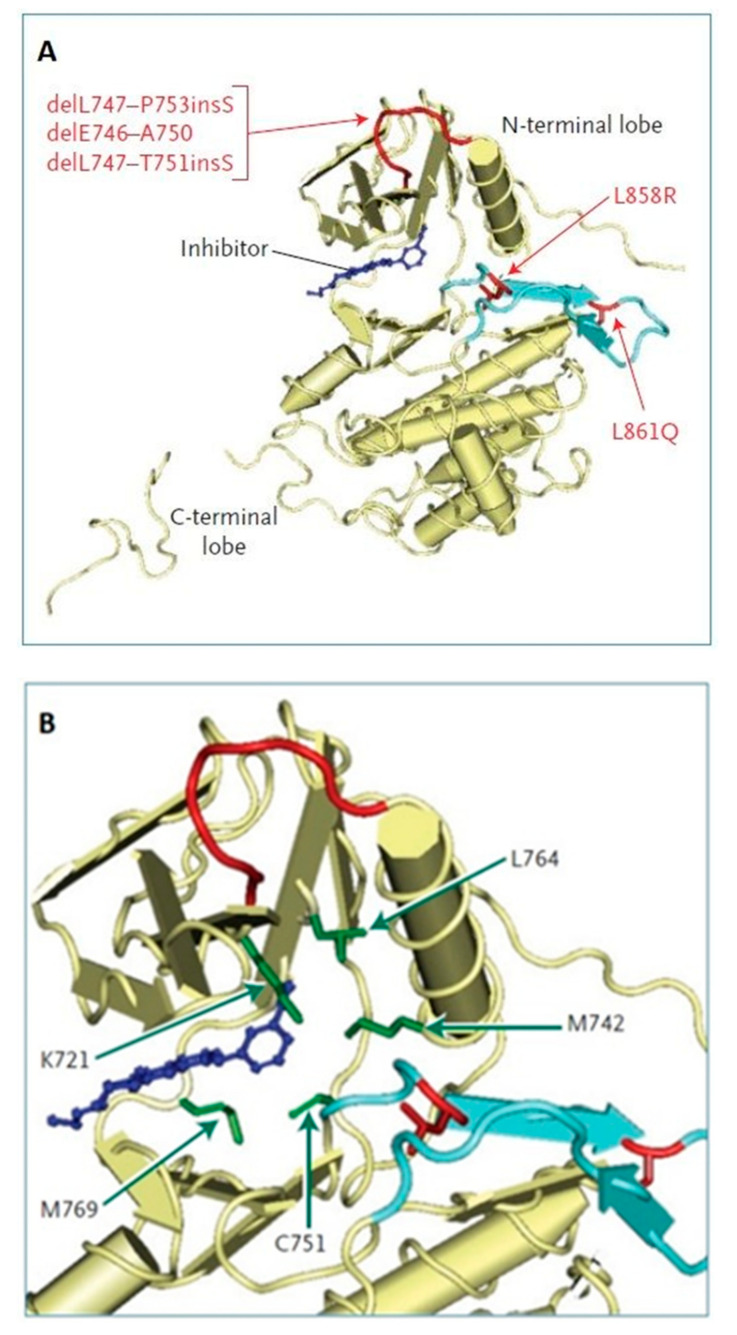
(**A**,**B**) Clustering of mutations in the Epidermal Growth Factor Receptor (EGFR) gene (adapted from Lynch T.J. et al., The New England Journal of Medicine, 2004) [23].

**Figure 4 ijms-22-00593-f004:**
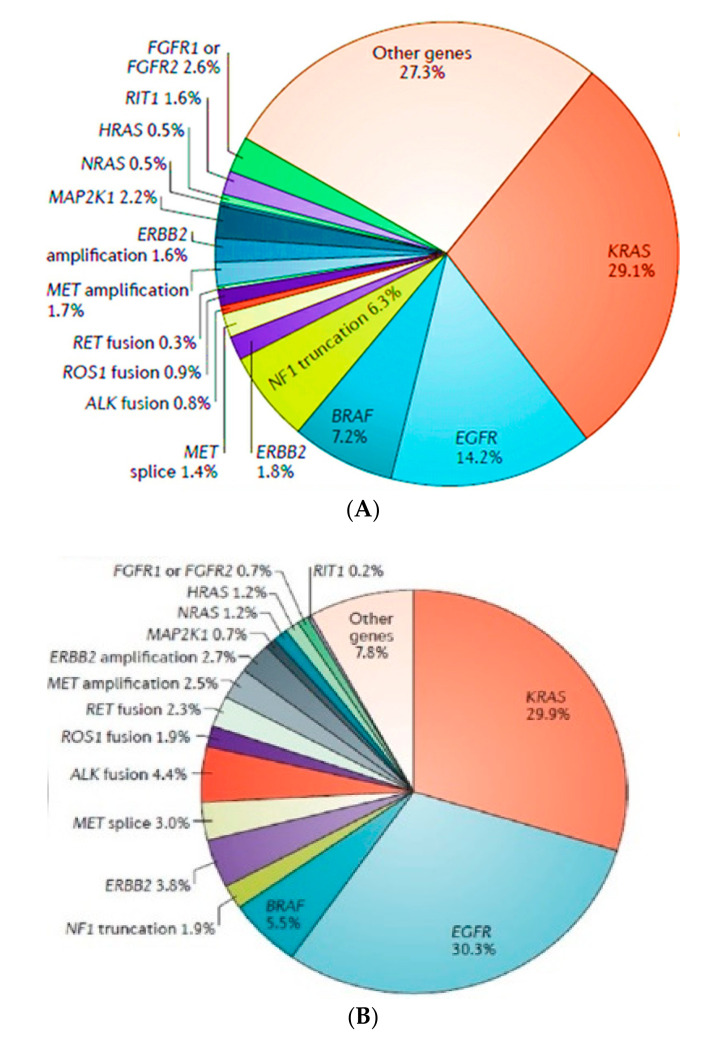
Distribution of the oncogenic driver mutations in non-small cell lung carcinoma (NSCLC) (adapted from Skoulidis F et al., 2019). (**A**) Early stage and (**B**) metastatic disease.

**Figure 5 ijms-22-00593-f005:**
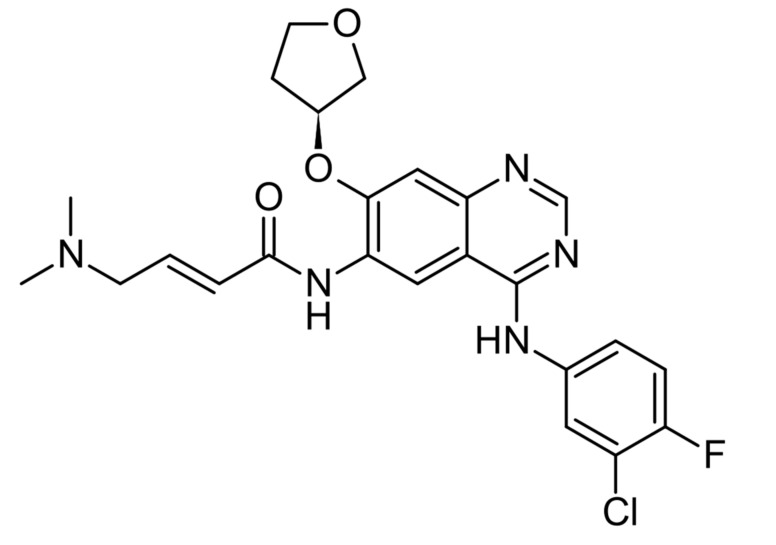
Afatinib—second-generation tyrosine kinase inhibitor.

**Figure 6 ijms-22-00593-f006:**
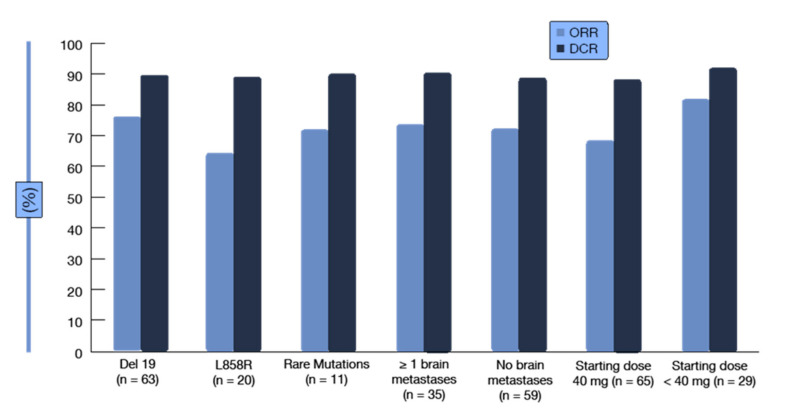
Overall response rates and disease control rates obtained with first-line afatinib in the noninterventional GIDEON study [59] (adapted from Brueckl et al., ESMO, 2018).

**Figure 7 ijms-22-00593-f007:**
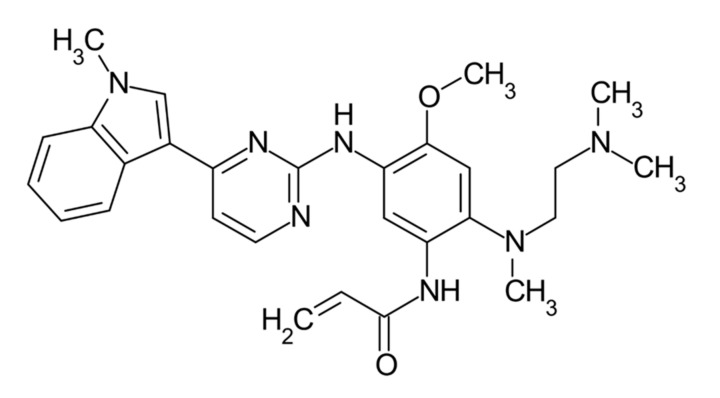
Osimertinib—third-generation tyrosine kinase inhibitor.

**Figure 8 ijms-22-00593-f008:**
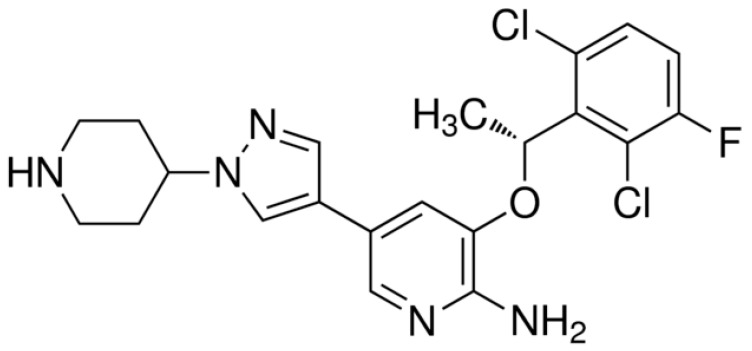
Crizotinib—first-generation Anaplastic Lymphoma Kinase tyrosine kinase inhibitor (ALK TKI).

**Figure 9 ijms-22-00593-f009:**
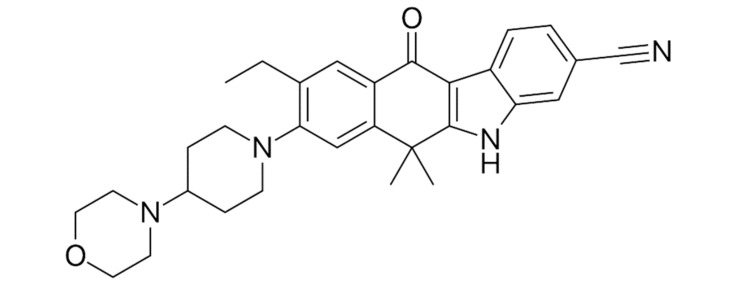
Alectinib—a second-generation ALK TKI.

**Figure 10 ijms-22-00593-f010:**
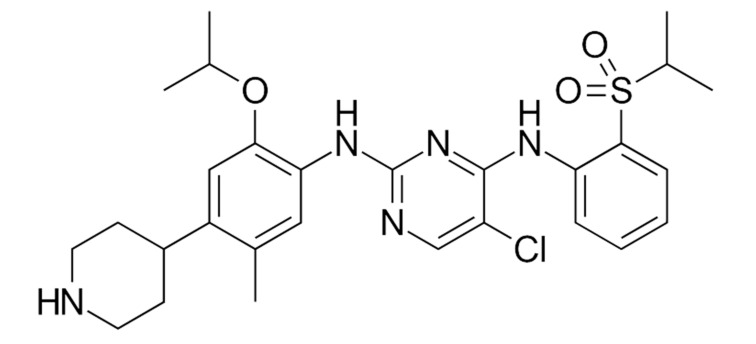
Ceritinib—a second-generation ALK TKI.

**Figure 11 ijms-22-00593-f011:**
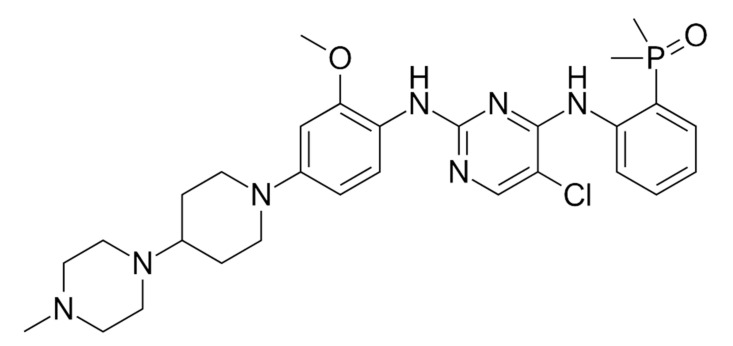
Brigatinib—a second-generation ALK TKI.

**Figure 12 ijms-22-00593-f012:**
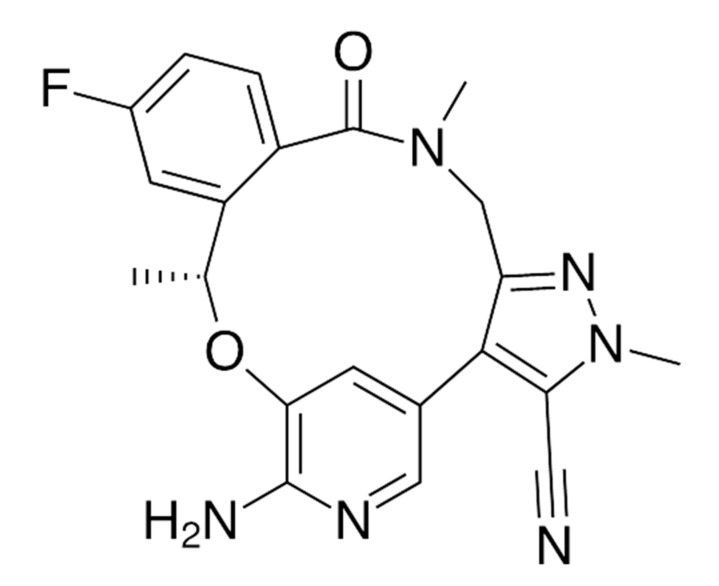
Lorlatinib—a third-generation ALK TKI.

**Table 1 ijms-22-00593-t001:** Prognostic stages in patients with central nervous system metastases according to the Radiation Therapy Oncology Group (RPA). KPS: Karnofsky Performance Status.

Prognostic Class	Characteristic	Median Survival (Months)
I	KPS ≥ 70, < 65 years, controlled primary tumor and no extracranial metastases	7.1
II	KPS ≥ 70, primary tumor not controlled	4.2
	KPS ≥ 70, controlled primary tumor ≥ 65 years	
	KPS ≥ 70, controlled primary tumor	
	<65 years and extracranial metastases	
III	KPS < 70	2.3

**Table 2 ijms-22-00593-t002:** Lung-molGPA (Lung Cancer Molecular Markers Graded Prognostic Assessment).

Prognostic Factor	Age (Years)	KPS	Extracranial Metastases	Number of BM	Gene Status
0	≥70	<70	Present	>4	EGFR neg/unk and ALK neg/unk
0.5	<70	70–80	-	1–4	NA
1	-	90–100	Absent	NA	EGFR-pos or ALK-pos

KPS—Karnofsky Performance Status, NA—not applicable, neg/unk—negative or unknown, pos—positive, BM—brain metastases, EGFR—Epidermal Growth Factor Receptor, and ALK—Anaplastic Lymphoma Kinase.

**Table 3 ijms-22-00593-t003:** Concentrations of the EGFR and ALK tyrosine kinase inhibitors in the cerebrospinal fluid (CSF).

Compound	CSF Penetration Rate (%)	CSF Concentration ng/mL or nM/L
Gefitinib	1.13 ± 0.36%	3.7 ± 1.9 ng/mL
		8.2 ± 4.3 nM/L
Erlotinib	2.8–5.1%	28.7 ± 16.8 ng/mL
		66.9 ± 39.0 nM/L
Afatynib	<1%	0.464 ng/mL
Crizotinib	0.26%	0.616 ng/mL
Alectinib	0.86	2.69 nM/L
Ceritinib	0.15	not reported
Lorlatinib	20–30%	not reported

## Data Availability

Not applicable.

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
