# Peer review of "Treatment of Brain Metastases of Non-Small Cell Lung Carcinoma"

_ijms, 2021, doi:10.3390/ijms22020593_

Round 1
Reviewer 1 Report
The Review article of Dr Rybarczyk-Kasiuchnicz et al aims to summarize the current state of the treatment of brain metastases of Non-Small-Cell Lung Carcinoma (NSCLCs). After a brief and didactic introduction of lung cancers the Authors present therapeutic options for patients. This manuscript provides an extensive analysis of the literature concerning tyrosine kinase inhibitors and ongoing clinical trials of TKI therapy; simultaneous CNS radiotherapy and TKI therapy as well as immunotherapy. This manuscript is well written and illustrated. This review would make therefore a suitable contribution for IJMS.
Minor
Abstract is too long and it will be preferable to be shortened (some considerations can be replaced in the Introduction and in the corresponding sections). Since the manuscript mainly deals with the development of targeting therapies in the treatment of NSCLCs, this could be mentioned as complement of the title or as a subtitle. In this connection, the Authors could also mention the conventional chemotherapy in use in the abstract.
Keywords are missing.
Line 65 and Figure 1, the Authors mentioned “Metastatic cells arrest at distinct sites and extravasate through vascular walls into the brain parenchyma.” and “vascular arrest by size exclusion”. Does the “seed and soil” hypothesis, - brain-derived signals promote the adhesion of cancer cells to intracranial blood vessels and foster metastasis formation, - actually ruled out?
Table 1, prognostic Class II, please suppress the “.” In front of “<65 years”.
Abbreviation of KPS need to be explained and introduced here and not in Table 2. For the Readers, that are not physicians, the Authors could provide few information concerning Karnofsky index.
Lines 154 and 156: Legend to Figure 2 should be corrected and the panels must be named A and B.
Figure 4. The labeling “a” and “b” in front of the pie charts is missing. The KRAS and BRAF mutations seems to be highly frequent in NSCLSs at early stage and in the metastatic disease. Do these mutations impact the response to the EGFR inhibitors? In the case of colorectal cancers, RAS mutations are associated to refractoriness to anti-EGFR antibodies. In this case, does the genotyping of NSCLCs routinely performed? What about HER2 and MET overexpression, and other signaling pathways such as PI3K/PTEN/AKT pathway?
Figure 9 is missing.
In general, the text of manuscript needs to be revised in view of misprints, for example:
Line 299 and 418, Please, change “erlotynib” to “erlotinib” …
Line 314, Please, change “non-smell” to “non-small”
To summarize the Review, the Authors could provide the decision chart for the management of patients with Non-Small-Cell Lung Carcinoma, taking into account the clinical characteristics of the tumors (number of metastatic lesions in the CNS, extracranial metastases), their biological characteristics (presence/absence of EGFR/ ALK alteration), the patient general health status, the treatments proposed and the prognosis. Such diagram could serve as a graphical abstract.
Author Response
WAS 3B. shows a close-up view of the EGFR tyrosine kinase domain, with the critical amino acids 182 implicated in binding the inhibitor. Specifically, 4-anilinoquinazoline compounds such as gefitinib 183 inhibit catalysis by occupying the ATP-binding site, where they form hydrogen bonds with 184 methionine769 (M769) and cysteine751 (C751) residues, whereas their anilino ring is close to 185 methionine742 (M742), lysine721 (K721), and leucine764 (L764) residues (all shown in green).In186 frame deletions within the loop that is targeted by mutations (shown in red) are predicted to alter the 187 position of these amino acids relative to that of the inhibitor. Mutated residues (red) are shown within 188 the activation loop of the tyrosine kinase (light blue). Gain-of-function mutations [35] in the tyrosine 189 kinase domain of the EGFR gene markedly increase sensitivity to EGFR TKIs. It has been shown that 190 10-30% of all lung adenocarcinomas [38] contain an EGFR activating mutation. EGFR mutations occur 191 mostly in adenocarcinoma, younger women and girls [27], and never-smokers [24,25,26]. The 192 increased prevalence [35] of EGFR mutations in the metastatic disease (early stage-14,2%, metastatic193 30,3%) – the dataset may partially reflect referral bias [27] (Figure 4).
DONE
1.From line 181 to line 188 there is a description of figure 3b
This is the description in the manuscript:
3B. shows a close-up view of the EGFR tyrosine kinase domain, with the critical amino acids 182 implicated in binding the inhibitor. Specifically, 4-anilinoquinazoline compounds such as gefitinib 183 inhibit catalysis by occupying the ATP-binding site, where they form hydrogen bonds with 184 methionine769 (M769) and cysteine751 (C751) residues, whereas their anilino ring is close to 185 methionine742 (M742), lysine721 (K721), and leucine764 (L764) residues (all shown in green).In186 frame deletions within the loop that is targeted by mutations (shown in red) are predicted to alter the 187 position of these amino acids relative to that of the inhibitor. Mutated residues (red) are shown within 188 the activation loop of the tyrosine kinase (light blue).
2.From the middle of verse 188 there is the rest of the manuscript text
Gain-of-function mutations [35] in the tyrosine 189 kinase domain of the EGFR gene markedly increase sensitivity to EGFR TKIs. It has been shown that 190 10-30% of all lung adenocarcinomas [38] contain an EGFR activating mutation. EGFR mutations occur 191 mostly in adenocarcinoma, younger women and girls [27], and never-smokers [24,25,26]. The 192 increased prevalence [35] of EGFR mutations in the metastatic disease (early stage-14,2%, metastatic193 30,3%) – the dataset may partially reflect referral bias [27]
- In the line 277 is no footnote (in bibliography no 70).The proper body of the sentence is as follows:
In the AURA 3 clinical trial [70], it was compared to pemetrexed and cisplatin or carboplatin-based, two-drug chemotherapy [65,66]. DONE
4.In the line 303 is no footnote (in bibliography no 72).The propter body of the sentence is as follows:
It is also effective in patients with metastases to the central nervous system and the meninges [72,74,75]. DONE
- In the line 342 the chemical formula for alectinib in missing
(lacks of the chemical formula of alectinib)
Figure 9. Alectinib - a second-generation ALK TKI
The chemical formula- DONE

Reviewer 2 Report
The Review article authored by of Dr Rybarczyk-Kasiuchnicz et al is in line with previous original works on lung cancer from this group. A previous review from this Team published in 2018 concerned the targeted therapy of lung cancer. The present report is focused on the treatment of brain metastases of non-small cell lung carcinomas using similar therapeutical approach. The Review successively describes the treatment of patients with wild-type EGFR and ALK lung cancers, followed by the improvement of the management of patients with lung cancer bearing a mutant EGFR or ALK using selective tyrosine kinase inhibitors (TKIs), and further describes the different generations of TKIs. Then, the Authors describes the immunotherapy trials on patients with non-small cell lung carcinomas.
The manuscript is pleasant to read and comprehensible. The clinical trials and their efficiency are detailed and clearly described, the acquired tumor resistance, the alternative therapeutic options and the responses are clearly exposed. This manuscript fits perfectly the topic of the Special Issue of International Journal of Molecular Sciences "Targeted Cancer Therapy and Mechanisms of Resistance” and I consider that it is worth publishing in this Special Issue.
Minor points
The Abstract is rather long and probably exceeds the recommended length.
On the contrary, the Introduction section is very short. It might be interesting to introduce for the reader -who might not be a specialist in lung cancer- some clues on the different types of lung cancers (SCLC; NSCLC, large cell, carcinoma, squamous cancers,…), their relative incidence, their etiology and risk factors, and their prognosis. The Author might provide an additional illustration (e.g. pie chart).
The Authors might also propose a conclusion/ perspective for this manuscript. It might concern the next challenges in the development/ improvement of therapies: e.g. the identification of novel molecular targets; the cross-talk of signaling pathways/feed-back control that may impact tumor responsiveness; the selective role of the stroma; or the development of alternative approaches (e.g. nanodevices) to enhance blood–brain barrier penetration of active compounds. Alternatively, the Authors might place brain tumors in an overall context and might discuss the similarities and specificities of lung metastases toward brain metastases from other cancers (breast and colon cancers, melanomas) -or brain primary tumors-, and whether there has been join benefit and/or fruitful clinical transfer from one pathology to the others.
The structure of Alectinib (Figure 9) has seemingly been lost during the edition of the manuscript.
Author Response

(The authors gave the same response as above.)
